# Electromagnetic Inverse Scattering from a Single Transmitter

## Abstract

Electromagnetic Inverse Scattering Problems (EISP) seek to reconstruct relative permittivity from scattered fields and are fundamental to applications like medical imaging. This inverse process is inherently ill-posed and highly nonlinear, making it particularly challenging, especially under sparse transmitter setups, *e.g.*, with only one transmitter. While recent machine learning-based approaches have shown promising results, they often rely on time-consuming, case-specific optimization and perform poorly under sparse transmitter setups. To address these limitations, we revisit EISP from a data-driven perspective. The scarcity of transmitters leads to an insufficient amount of measured data, which fails to capture adequate physical information for stable inversion. Accordingly, we propose a *fully end-to-end* and *data-driven* framework that predicts the relative permittivity of scatterers from measured fields, leveraging data distribution priors to compensate for the incomplete information from sparse measurements. This design enables data-driven training and *feed-forward* prediction of relative permittivity while maintaining strong robustness to transmitter sparsity. Extensive experiments show that our method outperforms state-of-the-art approaches in reconstruction accuracy and robustness. Notably, we demonstrate, for the first time, high-quality reconstruction from a single transmitter. This work advances practical electromagnetic imaging by providing a new, cost-effective paradigm to inverse scattering.

## 1 Introduction

Electromagnetic waves can penetrate object surfaces, making them essential for non-invasive imaging (Geng et al., 2024; O'Loughlin et al., 2018). At the core of electromagnetic imaging lies the Electromagnetic Inverse Scattering Problems (EISP), which seeks to reconstruct an object's relative permittivity from measured scattered electromagnetic field (Nikolova, 2011). By solving EISP, we can accurately recover internal structures without physical intrusion (Song et al., 2005), enabling a range of scientific and industrial applications, such as safer and more cost-effective alternatives to X-rays and MRI scans (Bevacqua et al., 2021; O'Loughlin et al., 2018; Nikolova, 2011). Typically, EISP necessitate a large number of transmitters and receivers to acquire sufficient measurement data. This requirement, however, leads to increased operational time and higher costs, thereby limiting the practical applicability of electromagnetic imaging techniques (Leigsnering et al., 2011). In contrast, reducing the number of transmitters offers significant advantages, including lower costs and easier deployment in constrained environments.(Baraniuk and Steeghs, 2007; Anitori et al., 2010)

However, the inherent ill-posed nature of EISP poses significant challenges to accurate reconstruction (Pan et al., 2011; Chen, 2018; Li et al., 2019; Zhong et al., 2016; Luo et al., 2024), particularly when only a limited number of transmitters are available. The scarcity of transmitters leads to an insufficient amount of measured data, which fails to capture adequate physical information for stable inversion. As a result, approaches relying solely on physical mechanisms(Slaney et al., 1984; Belkebir et al., 2005; Chen, 2009; Zhong and Chen, 2011) often fail to achieve accurate reconstruction. Conventional numerical methods such as backpropagation (BP) (Belkebir et al., 2005), generally fail to produce reliable reconstructions under such limited-data conditions. Recent machine learning-based approaches like PGAN (Song et al., 2021) and Physics-Net (Liu et al., 2022) often start with an initial solution derived from numerical methods, *i.e.*, BP, and frame the problem as an image-to-image translation task. With only a limited number of transmitters available, reliance on BP

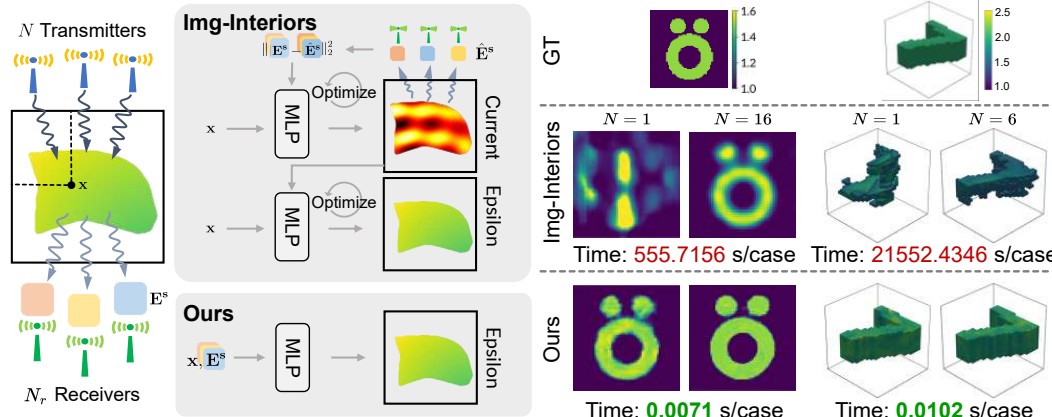

Figure 1: **Comparison between our method and the previous state-of-the-art. Left:** Img-Interiors (Luo et al., 2024) requires case-specific optimization to reconstruct the permittivity. In contrast, our method is a data-driven framework that operates in an *end-to-end*, *feed-forward* manner for solving inverse scattering. **Right:** Our method yields more accurate reconstructions than Img-Interiors (Luo et al., 2024). It remains robust even with a *single* transmitter and achieves real-time inference with over $70,000\times$ speed-up.

becomes a critical bottleneck. When BP fails, these methods are unable to correct its errors, as they are not fully end-to-end, ultimately leading to inaccurate reconstructions. The most recent method Img-Interiors (Luo et al., 2024) integrates physical mechanisms into neural networks and performs case-by-case optimization. However, in limited-transmitter scenarios, even after optimization has converged, the resulting reconstructions may still diverge substantially from the ground truth (Fig. 2), underscoring the intrinsic ambiguity of the inverse problem.

To address these limitations, we propose a *fully end-to-end* and *data-driven* framework that predicts the relative permittivity of scatterers from measured fields, leveraging data distribution priors to compensate for the scarcity of observational data. Specifically, our model takes the measured fields and the spatial coordinate of a position as input and directly predicts the relative permittivity at that location using Multilayer Perceptron (MLP)s, and is trained in a fully end-to-end manner against the ground-truth data. Our approach bypasses traditional numerical methods like BP, thereby avoiding the inherent constraints associated with conventional inversion techniques in limited-transmitter scenarios and fully exploiting the advantages of data-driven learning. This simple yet effective design enables efficient training across datasets and supports fast, feed-forward inference to achieve accurate and stable reconstruction predictions.

Extensive experiments demonstrate that our method outperforms existing State-of-the-Art (SOTA) methods on multiple benchmark datasets, especially under the challenging single-transmitter setting, where all previous methods fail (Fig. 5). It generalizes well to diverse scenarios and can be naturally extended to 3D scenes while maintaining high reconstruction accuracy. In summary, our contributions are threefold:

1) We systematically analyze the difficulty of lacking physical information faced by EISP in the setting of few transmitters, and point out that the missing information can be supplemented by data distribution priors.

2) Based on our analysis, we propose a *fully end-to-end* and *data-driven* model that does not rely on traditional numerical methods.

3) Extensive experiments show that our method outperforms existing SOTA approaches, especially under the challenging single-transmitter setting, marking a concrete step toward cost-effective and practical electromagnetic imaging solutions.

## 2 RELATED WORK

### 2.1 ELECTROMAGNETIC INVERSE SCATTERING PROBLEMS (EISP)

Solving EISP is to determine the relative permittivity of the scatterers based on the scattered field measured by the receivers, thereby obtaining internal imaging of the object. The primary challenges of EISP arise from its nonlinearity, ill-posedness, and errors introduced by the discretization (Pan et al., 2011; Chen, 2018; Li et al., 2019; Zhong et al., 2016; Luo et al., 2024). Traditional methods for solving EISP can be categorized into non-iterative (Slaney et al., 1984; Devaney, 1981; Habashy et al., 1993; Belkebir et al., 2005) and iterative (Chen, 2009; Zhong and Chen, 2011; Xu et al., 2017; Habashy et al., 1994; van den Berg et al., 1999) approaches. Non-iterative methods, such as the Born approximation (Slaney et al., 1984), the Eytov approximation (Devaney, 1981; Habashy et al., 1993), and the BP method (Belkebir et al., 2005), solve nonlinear equations through linear approximations, which inevitably lead to poor quality of the results. For better reconstruction quality, iterative methods (Zhong and Chen, 2009; Chen, 2009; Zhong and Chen, 2011; Xu et al., 2017; Habashy et al., 1994; van den Berg et al., 1999; Gao et al., 2015) such as 2-fold Subspace Optimization Method (SOM) (Zhong and Chen, 2009) and Gs SOM (Chen, 2009) are proposed. To further overcome the ill-posedness of EISP, diverse regularization approaches and prior information have been widely applied (Oliveri et al., 2017; Shen et al., 2014; Liu et al., 2018; Anselmi et al., 2018). However, all of these methods are not generalizable and can be time-consuming because of the iterative schemes (Liu et al., 2022).

### 2.2 MACHINE LEARNING FOR EISP

Recent studies shift to leverage neural networks to solve this problem and demonstrate promising results (Geng et al., 2021; Li et al., 2024). Some work (Wei and Chen, 2019; Li et al., 2019; Zhang et al., 2020; Xu et al., 2021; Sanghvi et al., 2019; Song et al., 2021; Liu et al., 2022) adopt a two-stage strategy: they use non-iterative methods such as BP (Belkebir et al., 2005) to generate initial estimates, which are then refined using image-to-image neural networks. While these approaches offer a degree of generalization, they are not end-to-end and remain dependent on BP initialization (Belkebir et al., 2005), which becomes their bottleneck. When physical data are too insufficient to reconstruct the scatterer, especially under single-transmitter settings, these approaches tend to "hallucinate" outputs according to unreliable initialization rather than predict the scatterer based on measured field (see Fig. 5). A more recent approach, Img-Interiors (Luo et al., 2024), integrates scattering mechanisms into the network architecture and achieves accurate reconstructions. However, it requires case-specific optimization, limiting generalization and making it vulnerable to local minima, often leading to failure in complex settings (see Figs. 4 and 6). Moreover, it fails under a single transmitter setting even when the optimization may have already converged because of ambiguity. While our method is also learning-based, it is an end-to-end feed-forward framework that simultaneously achieves generalization through data-driven learning. As a result, it consistently outperforms SOTA methods, particularly in the challenging single-transmitter setting where previous approaches fail.

## 3 REVISITING EISP

In this section, we revisit EISP and uncover its fundamental challenge: the inherent ill-posedness stemming from information scarcity.

**Preliminary.** In the forward process, the transmitters produce incident electromagnetic field $\mathbf{E}^i$ to the scatterer, generating scattered electromagnetic field $\mathbf{E}^s$. EISP is the inverse problem of the forward process. That is, for an unknown scatterer, we apply certain incident field $\mathbf{E}^i$ to it, and measure the scattered field $\mathbf{E}^s$ as our input. Our goal is to reconstruct the relative permittivity $\epsilon_r$ throughout the scatterer. For a detailed background introduction of the physical model of EISP, please refer to our supplementary material (Appx. B).

Specifically, the incident field $\mathbf{E}^i$ excites the induced current $\mathbf{J}$. Using the method of moments (Peterson et al., 1998), the total field $\mathbf{E}^t$ for a given transmitter can be expressed as (Colton and Kress, 2013):

$$\mathbf{E}^t = \mathbf{E}^i + \mathbf{G}^d \cdot \mathbf{J}, \qquad (1)$$

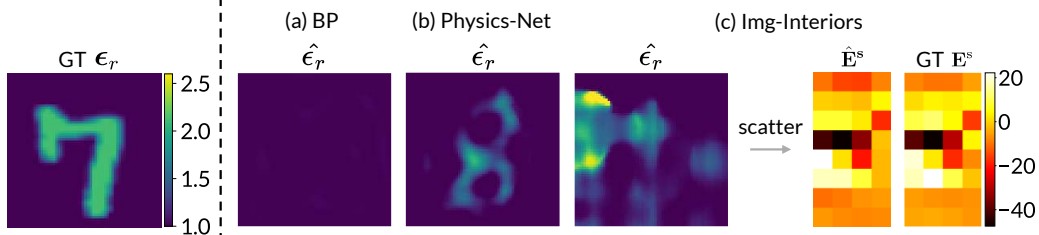

Figure 2: **Difficulties that previous methods faced under a single-transmitter setting.** (a) BP cannot reconstruct the scatterer. (b) Physics-Net makes incorrect guesses. (c) Although the reconstuction result of Img-Interiors is consistent with the measured field, the reconstructed scatterer itself is completely different from the ground truth.

where $\mathbf{E}^{\mathrm{t}}$ is a vector of length $M^2$, and $\mathbf{G}^{\mathrm{d}}$ is a constant $M^2 \times M^2$ matrix representing the discrete free-space Green's function in $\mathcal{D}$. The induced current field $\mathbf{J}$ satisfies:

$$\mathbf{J} = \mathrm{Diag}(\boldsymbol{\xi}) \cdot \mathbf{E}^{\mathrm{t}}, \tag{2}$$

where $\boldsymbol{\xi} = \boldsymbol{\epsilon}_r - 1$, $\mathrm{Diag}(\boldsymbol{\xi})$ represents a diagonal matrix whose leading diagonal consists of $\boldsymbol{\xi}$. Then $\mathbf{J}$ serves as a new source to emit $\mathbf{E}^{\mathrm{s}}$. For $N_r$ receivers, the scattered field $\mathbf{E}^{\mathrm{s}}$ can be got through $\mathbf{E}^{\mathrm{s}} = \mathbf{G}^{\mathrm{s}} \cdot \mathbf{J}$, where $\mathbf{G}^{\mathrm{s}}$ is a constant $N_r \times M^2$ matrix representing the discrete Green's function. Since $N_r \ll M^2$ in practice, reconstructing the induced current $\mathbf{J}$ from the scattered field $\mathbf{E}^{\mathrm{s}}$ is ill-posed.

**Reduction of measured data.** EISP is fundamentally challenged by nonlinearity and ill-posedness, especially when the amount of measured data is significantly reduced, such as under single-transmitter settings. We divide previous work into three categories and systematically analyze the difficulties they faced under this setting. (a) Conventional numerical approaches, such as BP (Belkebir et al., 2005), employ linear approximations, which limit their reconstruction quality. As shown in Fig. 2, BP cannot even reconstruct a rough shape of the scatterer. (b) Machine learning methods based on conventional numerical approaches Song et al. (2021); Liu et al. (2022), such as Physics-Net. Although Physics-Net can leverage data-driven training to compensate for missing physical information, its strong dependency on BP initialization becomes a critical bottleneck. When BP fails, the model cannot correct the error of BP because it is not fully end-to-end, resulting inaccurate reconstructions, as shown in Fig. 2. (c) Machine learning methods based on implicit functions, such as Img-Interiors (Luo et al., 2024). Img-Interiors reconstructs a scatterer through case-by-case optimization. As shown in Fig. 2, we use the scatterer reconstructed by Img-Interiors to simulate the scattered field, and the result closely matches the measured field. However, the scatterer itself deviates significantly from the ground truth, which shows the intrinsic ambiguity of the inverse problem. The core conclusion is that the severe information deficit makes a direct solution to the inverse problem fundamentally intractable. Consequently, any such attempt is bound to be fragile, highlighting the need for an alternative paradigm.

## 4    METHOD

### 4.1    OVERVIEW

To address the aforementioned limitations, we introduce our end-to-end, data-driven framework for EISP, as illustrated in Fig. 3. Our method employs an MLP that takes space coordinates $\mathbf{x}$ and corresponding scattered field measurements $\mathbf{E}^{\mathrm{s}}$ as input, and directly outputs the relative permittivity $\boldsymbol{\epsilon}_r$ at the specified locations. This approach effectively learns the mapping between scattered field $\mathbf{E}^{\mathrm{s}}$ and relative permittivity $\boldsymbol{\epsilon}_r$ through training on diverse scattering scenarios, thereby incorporating essential data distribution priors to compensate for the lack of physical information caused by insufficient measurements.

In the following, we detail our model architecture (Sec. 4.2), and the training losses (Sec. 4.3).

### 4.2    MODEL ARCHITECTURE

Based on the forward formulation of EISP in Sec. 3, where the scattered field measurements serve as input and the relative permittivity distribution represents the output, we design an end-to-end learning

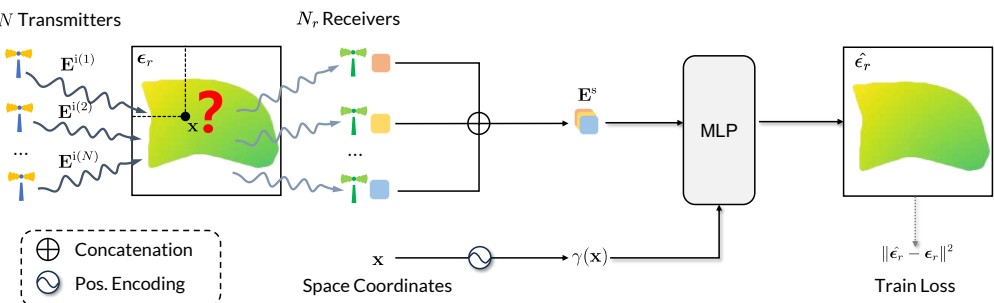

Figure 3: **Overview of our method.** Our pipeline is built around a MLP that serves as the inverse solver. Given the scattered field measurements $\mathbf{E}^s$ from all transmitters and receivers, along with a spatial query $\mathbf{x}$, the MLP directly predicts thee relative permittivity $\hat{\epsilon}_r(\mathbf{x})$. To enhance spatial expressiveness, we apply positional encoding $\gamma(\mathbf{x})$ to the query position. During training, dashed lines indicate the supervision signals applied.

framework that directly learns this complex nonlinear mapping. As illustrated in Fig. 3, our approach employs an MLP that approximates the inverse mapping from spatial coordinates and scattered field data to the relative permittivity values, formulated as:

$$\hat{\epsilon}_r(\mathbf{x_i}) = F_\theta(\mathbf{E}^s, \gamma(\mathbf{x_i})), \mathbf{x_i} \in \mathbb{R}^2, \tag{3}$$

where $\mathbf{x_i}$ represents the spatial coordinate, $\mathbf{E}^s$ denotes the scattered field measured by all receivers, $F_\theta(\cdot)$ is an MLP with trainable parameters, and $\hat{\epsilon}_r(\mathbf{x_i})$ is the predicted relative permittivity at the corresponding position. Recall that in Sec. 3, for a single transmitter, the scattered field $\mathbf{E}^s$ is discretized as a real-valued vector of dimension $2N_r$, containing the real and imaginary parts of the measurements from all $N_r$ receivers. In the multiple transmitter configuration, $\mathbf{E}^s$ is constructed by combining the complex measurement data from all $N$ transmitters, resulting in a real-valued vector of dimension $2N \cdot N_r$ that represents the wave propagation and scattering behavior under diverse illumination conditions provided by transmitters at different locations.

To enhance the model's capacity to represent high-frequency features, we apply positional encoding to the spatial coordinates $\mathbf{x_i}$, mapping them into a higher-dimensional Fourier feature space using the encoding function: $\gamma(x) = [\sin(x), \cos(x), \dots, \sin(2^{\Omega-1}x), \cos(2^{\Omega-1}x)]$, where the hyperparameter $\Omega$ controls the spectral bandwidth.

The complete relative permittivity distribution $\hat{\epsilon}_r$ is reconstructed by sampling the MLP at all grid points $\{\mathbf{x_i}\}_{i=1}^{M^2}$: $\hat{\epsilon}_r = \{F_\theta(\mathbf{E}^s, \gamma(\mathbf{x_i}))\}_{i=1}^{M^2}$.

### 4.3 TRAINING

Our training objective is defined by a single loss function designed to directly supervise the reconstruction accuracy of the relative permittivity distribution. The loss is formulated as: $\mathcal{L} = \|\hat{\epsilon}_r - \epsilon_r\|^2$. where $\hat{\epsilon}_r$ denotes the predicted relative permittivity and $\epsilon_r$ represents the ground truth. By minimizing this Mean Squared Error (MSE) loss between the predicted and true permittivity values, the model learns to infer the material properties directly from the scattered field measurements, effectively leveraging the data distribution priors to overcome the ill-posedness of the inverse problem. This simplified loss function ensures stable and efficient training.

## 5 EXPERIMENTS

### 5.1 SETUP

**Datasets.** We train and test our method on standard benchmarks used for EISP following previous work (Wei and Chen, 2019; Song et al., 2021; Liu et al., 2022). To enhance training efficiency and model robustness, datasets that share identical transmitter and receiver configurations are combined

Table 1: **Quantitative comparison results with SOTA methods.** For Circular and MNIST datasets, we report results under two noise levels: 5% and 30%. The best results are shown in **bold**, and the second-best results are underlined.

| Method | MNIST (5%) | | | MNIST (30%) | | | Circular (5%) | | | Circular (30%) | | | IF | | |
|---|---|---|---|---|---|---|---|---|---|---|---|---|---|---|---|
| | MSE ↓ | SSIM ↑ | PSNR ↑ | MSE ↓ | SSIM ↑ | PSNR ↑ | MSE ↓ | SSIM ↑ | PSNR ↑ | MSE ↓ | SSIM ↑ | PSNR ↑ | MSE ↓ | SSIM ↑ | PSNR ↑ |
| Number of Transmitters: $N = 16$ | | | | | | | | | | | | | $N = 8/18$ | | |
| BP (2005) | 0.177 | 0.719 | 16.43 | 0.178 | 0.716 | 16.38 | 0.052 | 0.905 | 27.41 | 0.053 | 0.904 | 27.42 | 0.190 | 0.779 | 16.19 |
| 2-fold SOM (2009) | 0.154 | 0.757 | 20.93 | 0.156 | 0.738 | 20.84 | 0.031 | 0.917 | 32.23 | 0.038 | 0.889 | 30.63 | - | - | - |
| Gs SOM (2009) | 0.072 | 0.923 | 28.31 | 0.081 | 0.901 | 27.13 | 0.023 | 0.946 | 35.40 | **0.024** | 0.937 | 34.89 | 0.184 | 0.790 | 17.00 |
| BPS (2019) | 0.093 | 0.909 | 25.00 | 0.105 | 0.891 | 23.90 | 0.027 | 0.963 | 33.00 | 0.029 | **0.956** | 32.42 | 0.310 | 0.664 | 17.05 |
| PGAN (2021) | 0.084 | 0.916 | 25.80 | 0.091 | 0.910 | 25.31 | 0.026 | **0.966** | 35.56 | 0.032 | 0.947 | 33.91 | 0.121 | **0.926** | 24.78 |
| Physics-Net (2022) | 0.075 | 0.932 | 26.17 | 0.093 | 0.906 | 24.58 | 0.027 | 0.934 | 32.72 | 0.030 | 0.927 | 32.08 | 0.170 | 0.788 | 18.48 |
| Img-Interiors (2024) | 0.200 | 0.863 | 26.41 | 0.336 | 0.760 | 19.01 | 0.036 | 0.947 | 35.05 | 0.047 | 0.932 | 32.62 | 0.153 | 0.837 | 23.26 |
| **Ours** | **0.039** | **0.978** | **32.11** | **0.050** | **0.966** | **29.91** | **0.020** | 0.965 | **36.92** | **0.024** | 0.954 | **35.19** | **0.094** | 0.916 | **24.89** |
| Number of Transmitters: $N = 1$ | | | | | | | | | | | | | | | |
| BP (2005) | 0.194 | 0.698 | 15.40 | 0.194 | 0.696 | 15.40 | 0.065 | 0.892 | 25.30 | 0.065 | 0.892 | 25.30 | 0.199 | 0.770 | 16.29 |
| 2-fold SOM (2009) | 0.432 | 0.556 | 12.49 | 0.828 | 0.382 | 9.45 | 0.060 | 0.859 | 24.61 | 0.157 | 0.639 | 20.07 | - | - | - |
| Gs SOM (2009) | 0.460 | 0.598 | 15.31 | 0.404 | 0.557 | 14.91 | 0.046 | 0.888 | 29.62 | 0.051 | 0.862 | 28.77 | 0.192 | 0.779 | 16.66 |
| BPS (2019) | 0.189 | 0.774 | 18.75 | 0.205 | 0.744 | 17.97 | 0.045 | 0.891 | 29.29 | 0.055 | 0.862 | 27.68 | 0.348 | 0.669 | 16.18 |
| PGAN (2021) | 0.133 | 0.867 | 21.69 | 0.153 | 0.830 | 20.41 | 0.033 | **0.932** | **32.02** | 0.040 | **0.914** | 29.94 | 0.248 | 0.680 | 16.85 |
| PhysicsNet (2022) | 0.137 | 0.798 | 19.98 | 0.152 | 0.783 | 19.38 | 0.055 | 0.887 | 26.60 | 0.056 | 0.890 | 26.48 | 0.175 | 0.771 | 17.45 |
| Img-Interiors (2024) | 0.305 | 0.604 | 16.06 | 0.467 | 0.484 | 12.47 | 0.096 | 0.855 | 26.19 | 0.153 | 0.806 | 20.90 | 0.305 | 0.705 | 17.34 |
| **Ours** | **0.085** | **0.921** | **26.09** | **0.127** | **0.862** | **22.56** | **0.031** | 0.931 | **33.18** | **0.038** | 0.914 | **31.38** | **0.128** | **0.908** | **24.19** |

into a unified training set. 1) Synthetic Circular-cylinder dataset (Circular) (Luo et al., 2024) is synthetically generated comprising images of cylinders with random relative radius, number, and location and permittivity. 2) Synthetic MNIST dataset (MNIST) (Deng, 2012) contains grayscale images of handwritten digits. For the two synthetic datasets, we follow previous work(Luo et al., 2024; Deng, 2012), we evaluate two levels of noise: 5% and 30%. 3) Real-world Institut Fresnel's database (IF) (Geffrin et al., 2005) contains three different dielectric scenarios, namely FoamDielExt, FoamDielInt, and FoamTwinDiel. 4) Synthetic 3D MNIST dataset (3D MNIST) (de la Iglesia Castro) contains 3D data of handwritten digits. 5) 3D ShapeNet dataset (3D ShapeNet) (Wu et al., 2015) contains 3D data of various shapes. For more details about datasets, please refer to Appx. C.

**Baselines and Metrics.** We maintain the same settings as in previous studies (Wei and Chen, 2019; Song et al., 2021; Sanghvi et al., 2019) to ensure a fair comparison. We compare our method with three traditional methods and four deep learning-based approaches: 1) **BP** (Belkebir et al., 2005): A traditional non-iterative inversion algorithm. 2) **2-fold SOM** (Zhong and Chen, 2009): A traditional iterative minimization scheme by using SVD decomposition. 3) **Gs SOM** (Chen, 2009): A traditional subspace-based optimization method by decomposing the operator of Green's function. 4) **BPS** (Wei and Chen, 2019): A CNN-based image translation method with an initial guess from the BP algorithm. 5) **Physics-Net** (Liu et al., 2022): A CNN-based approach that incorporates physical phenomena during training. 6) **PGAN** (Song et al., 2021): A CNN-based approach using a generative adversarial network. 7) **Img-Interiors** (Liu et al., 2022): An implicit approach optimized by forward calculation. Following previous work (Liu et al., 2022), we evaluate the quantitative performance of our method using PSNR (Wang and Bovik, 2009), SSIM (Wang et al., 2004), and Relative Root-Mean-Square Error (MSE) (Song et al., 2021).

## 5.2 COMPARISON WITH SOTAs

### 5.2.1 MULTIPLE TRANSMITTER EVALUATION

We begin by comparing our method against prior approaches under the multiple-transmitter setting, using both synthetic and real datasets for comprehensive evaluation. As shown in the upper part of Tab. 1, our method achieves comparable or superior performance to the SOTA in most cases, demonstrating how our end-to-end training framework successfully leverage the data prior across diverse data domains.

In addition, we present a qualitative comparison, as shown in Fig. 4. Traditional methods such as BP(Belkebir et al., 2005), Gs SOM (Chen, 2009), and 2-fold SOM (Chen, 2009) are only capable of recovering the coarse shape of the scatterer. BPS (Chen, 2018) produces sharp edges, but the reconstructed shapes are often inaccurate. PGAN (Song et al., 2021) achieves accurate shape

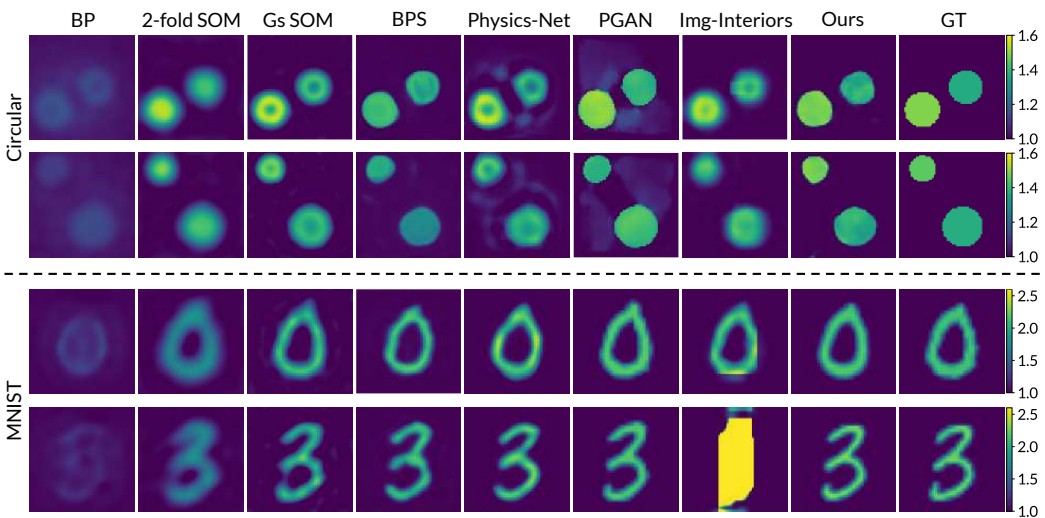

Figure 4: **Qualitative comparison under the multiple-transmitter setting.** The results are obtained with $N = 16$ transmitters and a noise level of 5%. Colors represent the values of the relative permittivity.

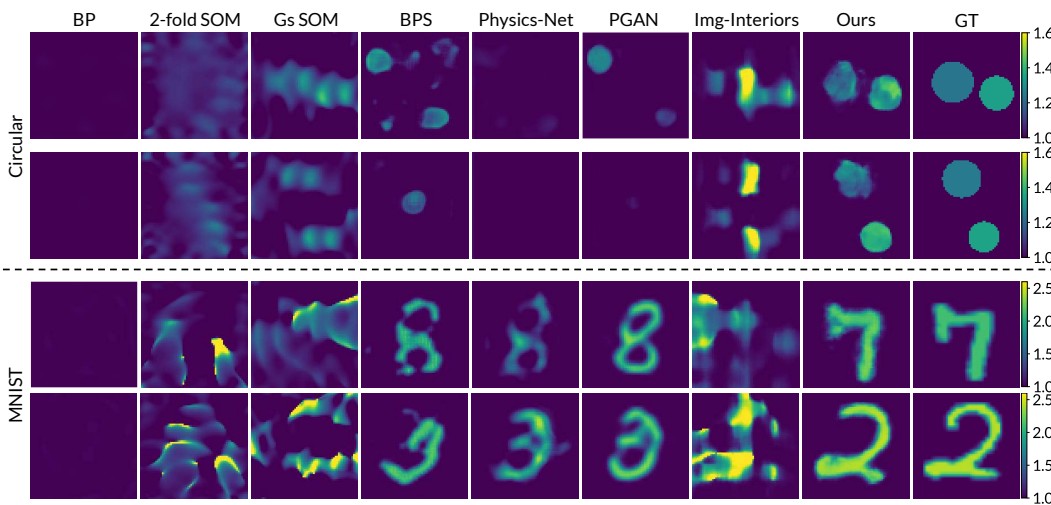

Figure 5: **Qualitative comparison under the single-transmitter setting.** Results are obtained with $N = 1$ transmitter and noise level of 5%. Colors represent the values of the relative permittivity.

recovery, yet introduces noticeable background artifacts. Img-Interiors (Luo et al., 2024) can generate high-quality reconstructions, but occasionally fails due to local optima, as it is based on an iterative optimization process (see the last row). In contrast, our method produces accurate and clean reconstructions across all cases, demonstrating both visual fidelity and robustness.

### 5.2.2 SINGLE TRANSMITTER EVALUATION

Furthermore, we investigate a highly challenging and practically important setting that has been largely underexplored in previous work: performing EISP with a minimal number of transmitters. Specifically, we consider the most extreme case, using only a single transmitter. As shown in the lower part of Tab. 1, our method significantly outperforms all previous approaches across all datasets and noise levels. This remarkable performance under such constrained conditions underscores the efficacy of our end-to-end training framework, which successfully encodes and leverages rich data priors to achieve state-of-the-art results across diverse domains.

Table 2: **Ablation study of noise levels effects on MNIST under the multiple-transmitter setting.**

| Noise Level | MSE ↓ | SSIM↑ | PSNR↑ |
|---|---|---|---|
| 5% | 0.039 | 0.978 | 32.11 |
| 10% | 0.039 | 0.978 | 32.18 |
| 15% | 0.043 | 0.973 | 31.30 |
| 20% | 0.043 | 0.974 | 31.34 |
| 25% | 0.046 | 0.970 | 30.59 |
| 30% | 0.050 | 0.966 | 29.91 |

Table 3: **Ablation study on training data size under the mutiple-transmitter setting.** Noise levels (5% and 30%) in parentheses.

| Data Size | MNIST (5%) | | | MNIST (30%) | | |
|---|---|---|---|---|---|---|
| | MSE ↓ | SSIM↑ | PSNR↑ | MSE ↓ | SSIM↑ | PSNR↑ |
| 100% | 0.039 | 0.978 | 32.11 | 0.050 | 0.966 | 29.91 |
| 75% | 0.043 | 0.974 | 31.63 | 0.059 | 0.956 | 28.77 |
| 50% | 0.048 | 0.968 | 30.68 | 0.068 | 0.944 | 27.69 |
| 25% | 0.064 | 0.948 | 28.89 | 0.101 | 0.902 | 25.44 |

To better understand this phenomenon, we present qualitative comparisons in Fig. 5. Traditional methods such as BP(Belkebir et al., 2005), Gs SOM (Chen, 2009), and 2-fold SOM (Chen, 2009) produce only blurry reconstructions. Deep learning-based methods like BPS (Chen, 2018), Physics-Net (Liu et al., 2022), and PGAN (Song et al., 2021) tend to "hallucinate" the digit, resulting in wrong shape on the MNIST dataset. Img-Interiors (Luo et al., 2024) fails to capture the fundamental morphology of the scatterer, resulting in structurally inaccurate representations that deviate significantly from the ground truth. Among all the methods, only ours can still produce reasonably accurate reconstructions of the relative permittivity under such an extreme condition.

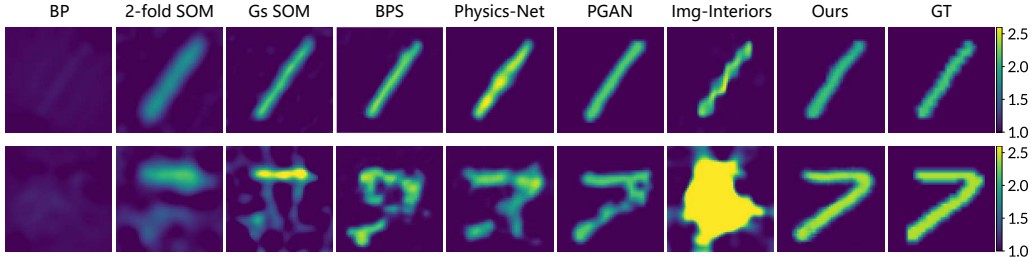

Figure 6: **Qualitative comparison under high noise setting.** The results are obtained with $N = 16$ transmitters and a noise level of 30%. Colors represent the values of the relative permittivity.

## 5.3 ABLATION STUDY

**Noise Robustness.** To simulate real-world sensor noise and related perturbations, we evaluate the robustness of the models by adding noise to the scattered field. Moving beyond simple binary testing, we systematically assess the model's performance across multiple noise levels ranging from 5% to 30% to examine its behavior in various noisy environments. The quantitative results presented in Tab. 2 demonstrate that our model exhibits smooth and gradual performance degradation as the noise level increases, maintaining excellent reconstruction capability even under strong noise interference as high as 30%. Qualitative visualizations in Fig. 6 show that most baseline methods exhibit noticeable artifacts or even complete failure under severe noise conditions, while our method remains robust and preserves the essential structure of the target.

**Ablation on Training Data Size.** To investigate the dependency of model performance on training data volume, we trained our model on varying scales of data from 100% down to 25% and evaluated them on a complete test set. The quantitative results are presented in Tab. 3. First, our model demonstrates remarkable data efficiency, maintaining strong performance even when trained on partial datasets. Second, the performance degradation becomes substantially more pronounced under high-noise conditions. The performance penalty for data reduction is markedly severer in high-noise scenarios. This pronounced contrast underscores that sufficient training data is crucial for the model to learn robust features capable of countering strong noise interference.

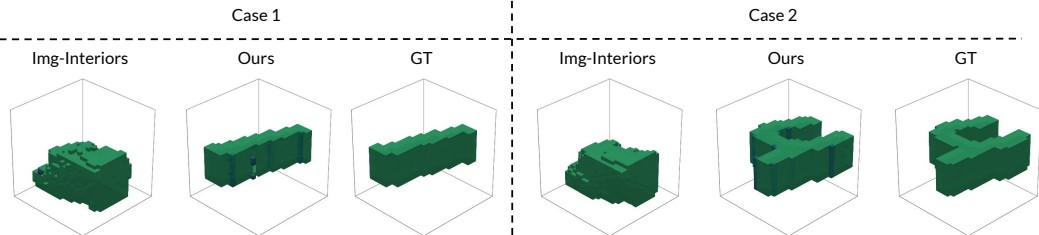

Figure 7: **Qualitative comparison under the single-transmitter setting for 3D reconstruction on 3D MNIST dataset.** The results are obtained with a single transmitter ($N = 1$). The voxel colors represent the values of the relative permittivity.

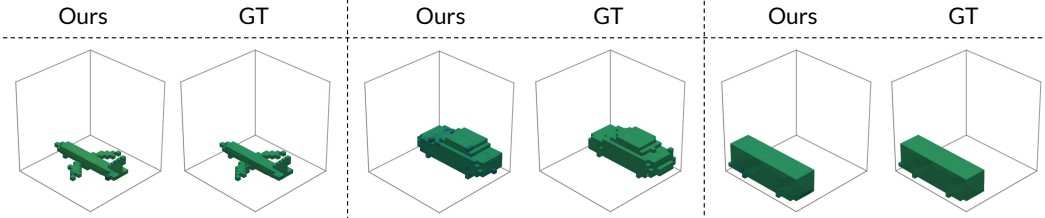

Figure 8: **Qualitative comparison under the single-transmitter setting for 3D reconstruction on 3D ShapeNet dataset.** The results are obtained with a single transmitter ($N = 1$). The voxel colors represent the values of the relative permittivity.

### 5.4 RECONSTRUCTION ON 3D DATA

**Setup and Metrics.** Our method can be naturally extended to 3D scenarios. Following previous work (Luo et al., 2024), we employ the Synthetic 3D MNIST (de la Iglesia Castro) and extend to 3D ShapeNet (Wu et al., 2015) for training and testing. For evaluation, we adopt 3D versions of the MSE (Song et al., 2021) and Intersection over Union (IoU) as our metrics. Further details on the datasets are provided in Appx. C.

**Results.** We evaluate our method and Img-Interiors under limited-transmitter settings. Quantitative results demonstrate the superiority of our approach: on 3D MNIST, our method achieves an MSE of 0.120 and IoU of 0.769 with $N = 1$ transmitter, significantly outperforming Img-Interiors which obtains an MSE of 0.372 and IoU of 0.094 under the same conditions. With $N = 6$ transmitters, our results further improve to MSE of 0.094 and IoU of 0.834. For the more complex 3D ShapeNet dataset under $N = 1$ configuration, our method obtains an MSE of 0.064 and IoU of 0.762, showcasing its generalization capability to diverse 3D structures. Fig. 1 provides a comprehensive comparison of reconstruction quality and runtime between the two methods for both $N = 1$ and $N = 6$ configurations. Fig. 7 and Fig. 8 provide visual comparisons of additional 3D reconstruction results on the 3DMNIST and 3DShapeNet datasets. These results show that our method maintains robustness and generalizes effectively across geometrically complex 3D structures, representing significant progress towards practical applications.

## 6 CONCLUSION

In this work, we propose a fully end-to-end data-driven framework for electromagnetic inverse scattering that directly predicts relative permittivity from scattered field measurements. By leveraging data distribution priors to compensate for the lack of physical information, our method demonstrates state-of-the-art reconstruction accuracy and robustness, particularly in challenging single-transmitter scenarios where existing methods fail. This work highlights the potential of data-driven approaches to overcome the ill-posedness of inverse problems and provides a practical path toward cost-effective electromagnetic imaging.

**Limitations.** While our method effectively handles sparse transmitter settings, it cannot deal with different locations of receivers or transmitters, which remains an important direction for future work.

## 7 STATEMENT

**Ethics statement.** This work does not involve any human subjects, or the use of sensitive data. Therefore, no ethical approval was required for this study.

**Reproducibility statement.** We are committed to ensuring the reproducibility of our results. The source code and all generated datasets used in this work will be released upon acceptance. Details of the data generation process, experimental setup, hardware specifications and full hyperparameter configurations are provided in Appx. C. To account for randomness, we report the mean and standard deviation of performance metrics across multiple runs with different random seeds, as shown in Appx. D.1. With the released code, data, and configuration files, all reported results can be reproduced within a small variance.

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
