# OpenReview forum: "Electromagnetic Inverse Scattering from a Single Transmitter"
_ICLR.cc/2026/Conference — ICLR 2026 Conference Withdrawn Submission_

### Official Review · Reviewer_sTLH · 2025-10-21

**Soundness:** 2
**Presentation:** 3
**Contribution:** 2
**Rating:** 4
**Confidence:** 3

**Summary:**

This paper addresses the electromagnetic inverse scattering problem (EISP) under the extremely sparse measurement condition — specifically, the single-transmitter setting. The authors propose a data-driven, end-to-end inverse solver based on a Multi-Layer Perceptron (MLP) that maps spatial coordinates and the measured scattered field to the local dielectric permittivity. The core claim is that data priors can compensate for missing measurement information, allowing stable reconstructions from only one transmitter. Experiments on multiple datasets show quantitative and qualitative improvements over several baselines. The authors also provide ablation studies on noise levels, data scale, and a 3D extension. Limitations are acknowledged — particularly, the method’s inability to generalize across varying transmitter/receiver positions.

**Strengths:**

(1)This approach addresses the challenging yet practical single-transmitter inverse scattering problem. The proposed end-to-end multilayer perceptron model replaces complex iterative solvers, significantly reducing inference time.

(2)Covers diverse datasets — Circular, MNIST, 3D MNIST, 3D ShapeNet, and real Fresnel measurements — with multiple noise levels and training data sizes, showing reasonable robustness.

(3)The paper is well-organized, and the writing is clear. The framework is clearly explained with helpful visualizations.

**Weaknesses:**

（1） The paper mentions joint training on multiple datasets, but it’s unclear whether baselines were trained under the same setup or reused from prior papers.

（2）While the paper emphasizes inference efficiency (e.g., much faster than Img-Interiors), it provides no quantitative analysis of the training cost, such as training time, GPU resources, dataset size, or convergence behavior. Since the proposed approach relies entirely on supervised learning with a large number of simulated examples, the offline training expense could be substantial and might offset the claimed efficiency advantage at deployment.

（3）The proposed method appears to be position-dependent — it assumes fixed transmitter and receiver positions during both training and inference. The authors even acknowledge in the Limitations section that the model cannot generalize to different sensor placements.
In practical electromagnetic imaging systems, however, sensor positions often vary slightly between measurements or need to be reconfigured for different objects, frequencies, or environments. If each new configuration requires full retraining from scratch, the method’s real-world applicability becomes severely constrained.

**Questions:**

（1）Were all baselines (Img-Interiors, Physics-Net, PGAN) retrained under identical datasets and conditions? If not, how is fairness ensured?

（2）It is necessary to supplement efficiency comparisons with other methods.  Without analyzing retraining overhead, the practicality of the method remains unclear.

（3）Real-world deployments demand flexible, recalibratable solvers that can adapt to new setups with minimal retraining. Without this, the method is more of a proof of concept than a deployable system.

（4）How does performance degrade when transmitter/receiver positions are slightly shifted? Please provide quantitative results.

（5）Are there any visualizations where the model fails (e.g., strong coupling or occlusion)? Analyzing such cases allows readers to better understand the effective boundaries of the method.

---

### Official Review · Reviewer_5JDx · 2025-10-30

**Soundness:** 3
**Presentation:** 3
**Contribution:** 2
**Rating:** 2
**Confidence:** 2

**Summary:**

This paper proposes a data-driven neural network approach for electromagnetic inverse scattering problems (EISP), where the goal is to reconstruct permittivity distributions from scattered field measurements. The method employs an MLP that takes scattered field measurements and spatial coordinates as input, using positional encoding to predict permittivity values at query locations. The authors demonstrate their approach on synthetic benchmarks including MNIST-based datasets and geometric shapes, as well as real-world data. The paper claims to achieve high-quality reconstruction even with a single transmitter, a severely ill-posed scenario.

**Strengths:**

S1: The paper provides a clear formulation of the electromagnetic inverse scattering problem and presents the methodology in an accessible manner. The problem setup, including the forward scattering model and the inverse reconstruction task, is well-explained for readers unfamiliar with this domain.

S2: The authors conduct experiments across multiple baseline methods. The visual comparisons demonstrate qualitative improvements over existing approaches in several scenarios.

S3: The paper extends the method to 3D reconstruction tasks, demonstrating that the approach can handle higher-dimensional problems beyond 2D cases, which shows some degree of scalability of the proposed framework.

**Weaknesses:**

W1: The core technical contribution of this paper is remarkably simple and lacks novelty. An MLP that takes scattered field measurements E^s and spatial coordinates x as input to predict permittivity, combined with standard positional encoding and a basic MSE loss on permittivity. The idea of training neural networks in a supervised, end-to-end fashion to learn direct mappings from input to desired outputs is a standard paradigm in machine learning and has been widely applied across various domains. The architectural components used are standard techniques borrowed from existing literature without any domain-specific innovation. The contribution appears to be primarily demonstrating that a standard supervised learning approach with sufficient training data can outperform existing methods on specific benchmarks, which is insufficient for a venue like ICLR that expects significant technical innovation.

W2: The paper claims their method "generalizes well to diverse scenarios" but provides no evidence of generalization beyond the training distribution, which is a critical flaw for a method claiming practical applicability. All experiments are in-distribution tests where training and test samples come from the same synthetic data generation process with identical physical parameters. Without out-of-distribution generalization experiments, we cannot assess whether the method is learning meaningful physical relationships or simply memorizing statistical patterns in the training data.

W3: There is no analysis of the ill-posedness of the problem. The authors claim the single-transmitter problem is severely ill-posed but then claim to solve it with data priors, without explaining what makes the problem well-posed under their framework or whether they are simply memorizing training data patterns.

W4: The paper makes strong claims about advancing "practical electromagnetic imaging" and providing a "cost-effective paradigm," but these claims are not supported by the evidence presented.

**Questions:**

Q1: The vast majority of experiments rely on synthetic data (MNIST digits and simple geometric shapes) which may not capture the complexity and variability of real electromagnetic scattering scenarios. Is this the common practice in this field?

Q2: The paper demonstrates that performance degrades with smaller training datasets (Table 3), but some information is missing. How many training samples are actually required to achieve the reported performance? What is the training time and computational resources needed for datasets of different sizes?

---

### Official Review · Reviewer_QtQX · 2025-10-31

**Soundness:** 2
**Presentation:** 2
**Contribution:** 1
**Rating:** 2
**Confidence:** 5

**Summary:**

The paper studies the electromagnetic inverse scattering problem (EISP) — reconstructing an object’s relative permittivity from measured scattered electromagnetic fields. Due to the limited data, solving such an inverse problem often suffers from the ill-posed issue. This work presents a fully end-to-end, data-driven framework that predicts permittivity directly from scattered fields using MLP networks.

**Strengths:**

+ This work shows superior performance over all tested baselines (BP, SOM, PGAN, Physics-Net, Img-Interiors) on synthetic (MNIST, Circular) and real (Institut Fresnel) datasets.
+ It also shows high-quality reconstructions with a single transmitter in the result section, under specific experimental settings.

**Weaknesses:**

The reviewer has a few major concerns regarding the weakness:
1. The main content of this work is applying existing simple ML models (MLP) to solve electromagnetic inverse problems, rather than developing new ML algorithms or theory. Therefore, this work is more suitable for publication in the field of computaitonal electromagnetics rather than in machine learning.
2. This paper claims that it address the ill-posedness issue of electromagnetic inverse problem. This contribution is overclaimed. While one can use a ML model to bypass solving the inverse problem (which has been studied in many papers), there is no theory to show that the predicted outcome has any accuracy gaurantees or physical meaning. The reviewer feels that this paper presented some cherry-picking results under specific experiemental setting, which is not enough to support the big claim of contribution.
3. There have been massive papers that apply deep learning to solve electromagnetic inverse problems. Many of these methods are not dsicussed or compared. Here is a short list (among many existing papers on google scholar):
(1) Yao, He Ming, E. I. Wei, and Lijun Jiang. "Two-step enhanced deep learning approach for electromagnetic inverse scattering problems." IEEE Antennas and Wireless Propagation Letters 18.11 (2019): 2254-2258.
(2) Wang, Ji-Yuan, and Xiao-Min Pan. "Universal Approximation Theorem and Deep Learning for the Solution of Frequency Domain Electromagnetic Scattering Problems." IEEE Transactions on Antennas and Propagation (2024).
(3) Yu, Xinling, et al. "PIFON-EPT: Mr-based electrical property tomography using physics-informed fourier networks." IEEE journal on multiscale and multiphysics computational techniques 9 (2023): 49-60.
(4) Yao, He Ming, Lijun Jiang, and E. I. Wei. "Enhanced deep learning approach based on the deep convolutional encoder–decoder architecture for electromagnetic inverse scattering problems." IEEE Antennas and Wireless Propagation Letters 19.7 (2020): 1211-1215.

**Questions:**

I have a few questions:
1. How should we place the transmittor to improve the prediction accuracy? How many data samples are needed to ensure good prediction accuracy?
2. While this method does not need to solve the inverse problem, how can this method gaurantee that the predicted solution is accurate and physically meaningful?

---

### Official Review · Reviewer_V6S8 · 2025-10-31

**Soundness:** 4
**Presentation:** 3
**Contribution:** 3
**Rating:** 4
**Confidence:** 3

**Summary:**

This paper tackles the challenging problem of Electromagnetic Inverse Scattering (EISP) from sparse data, particularly from a single transmitter. The authors argue that existing methods are limited by their reliance on traditional numerical methods (like BP), which fail with sparse data , or by time-consuming, case-specific optimization. This work proposes a fully end-to-end, data-driven framework that bypasses these traditional methods , using a Multilayer Perceptron (MLP) to directly predict relative permittivity from measured scattered fields and spatial coordinates. The model leverages data distribution priors to compensate for the lack of physical information from sparse measurements , enabling fast, feed-forward inference. The primary contribution is that the method outperforms existing SOTA approaches on several benchmarks and, for the first time, demonstrates high-quality, robust reconstruction in the highly challenging single-transmitter setting .

**Strengths:**

1. **Novel Problem Formulation:** This paper's primary contribution lies in its re-framing of the EISP problem. Instead of relying on traditional physics-based initializers (like BP) or slow, case-specific optimization, the authors propose a fully end-to-end, data-driven framework. This feed-forward model directly learns the mapping from scattered fields to permittivity by leveraging data distribution priors, which represents a clear paradigm shift from conventional approaches.

2. **Comprehensive Experimental Evaluation:** The authors conduct extensive experiments against a wide array of SOTA baselines (both traditional and learning-based) on multiple standard datasets, including synthetic and real-world data. The ablation studies on noise robustness and training data size provide valuable insights into the method's behavior under different conditions.

3. **Clear Presentation:** The paper is well-written and clearly structured. The authors effectively motivate the work by identifying the specific "bottleneck" of BP-based initialization and the ambiguity and speed limitations of optimization-based methods. The proposed method is presented in a straightforward manner, and the figures are illustrative.

**Weaknesses:**

1. **Unclear Methodological Innovation:** The methodology section is brief, making it difficult to identify the core technical contributions. From my reading, the approach appears to be a straightforward application of NeRF-like implicit neural representations to the electromagnetic imaging domain. Similar methods already exist in the literature (e.g., NeRF2: Neural Radio-Frequency Radiance Fields), which casts doubt on the novelty of this work. The authors need to clearly articulate what distinguishes their method from these existing approaches. What are the specific technical innovations beyond adapting an existing framework to a new application domain? Without this clarity, it is difficult to assess the paper's contribution to the field.

2. **Insufficient Validation on Real-World Data:** While the paper includes experiments on multiple datasets, the validation on real-world data is unconvincing. Only Table 1 provides quantitative results on the real-world dataset, but critically, there are no qualitative results (e.g., reconstructed images) shown for real data. This is a significant omission, as qualitative visualizations are essential for assessing reconstruction quality and identifying potential artifacts or failure modes. The current results do not clearly demonstrate the proposed method's advantages on real-world data, which severely limits the persuasiveness of the claims about practical applicability. Given that real-world performance is crucial for a method positioned as a practical solution for electromagnetic imaging, this lack of comprehensive real-data validation is a major weakness.

3. **Generalization Limitations:** The paper's most critical weakness is a fundamental limitation acknowledged by the authors: the model is trained for a fixed transmitter and receiver configuration and cannot generalize to different sensor locations. This severely restricts its practical deployment, where hardware setups often vary. For a method claiming to enable "practical electromagnetic imaging," this inflexibility is a major concern. The paper would be significantly strengthened if it proposed concrete solutions or at least thoroughly discussed potential paths to overcome this, such as conditioning the network on sensor coordinates.

If the authors can clearly articulate the methodological innovations (Weakness #1) and provide comprehensive qualitative results on real-world data that convincingly demonstrate the method's advantages (Weakness #2), I would consider revising my assessment upward.

**Questions:**

See Weaknesses

---

### Note · Authors · 2025-11-13

I have read and agree with the venue's withdrawal policy on behalf of myself and my co-authors.